# The Carbon Emissions Trading Policy of China: Does It Really Promote the Enterprises’ Green Technology Innovations?

**DOI:** 10.3390/ijerph192114325

**Published:** 2022-11-03

**Authors:** Xiaoqi Li, Dingfei Guo, Chao Feng

**Affiliations:** School of Economics and Business Administration, Chongqing University, Chongqing 400030, China

**Keywords:** carbon emission trading policy, green technology innovations, green invention patent applications, PSM−DID

## Abstract

The carbon emissions trading policy has profound impacts on the production and operation of enterprises. The aim of this study is to examine the effects of the carbon emissions trading policy on enterprises’ green technology innovations by using PSM−DID models. The results showed that: (1) the carbon emissions trading policy has a facilitating effect on green technology innovation of China’s enterprises in pilot cities; (2) there is significant spatial heterogeneity in this effect and it is extremely beneficial to enterprises’ green technology innovations in eastern China; and (3) the trading policy is proved to have significant positive effects on green technology innovations of non-state and non-high-tech enterprises, while it has no effects on that of state-owned and high-tech enterprises. The above findings were corroborated by the placebo test and other methods.

## 1. Introduction

With the industrial systems of modern society, human beings consume a large amount of fossil energy in terms of production and living, and at the same time emit a certain amount of carbon dioxide. The excessive increase in carbon emissions has led to a serious greenhouse effect, which poses a significant threat to global climate. Since 2006, China has transformed into a major carbon emitter, accounting for nearly 28.5% of global carbon emissions [1]. Recently, the Chinese government has committed to energy conservation and emission reduction. In 2015, the Political Bureau of the Central Committee of the Communist Party of China first proposed the concept of “Greenization” [2]. The government pledged to reduce carbon emissions intensity to 60–65% of 2005 levels by 2030 [3,4,5], and put forward a vision to “achieve peak emissions by 2030 and carbon neutrality by 2060”.

In order to achieve this goal as soon as possible, China approved a carbon emission trading mechanism policy in 2011. Carbon emissions trading originated from the Kyoto Protocol, which was adopted in December 1997. In the past, the state mainly controlled and restrained carbon emissions through administrative means, while market mechanisms were introduced after the launch of carbon trading. Carbon trading is a system of tradable allowances. Enterprises can use or trade carbon emissions based on the allowances, which are calculated from the pledged emission reductions listed in the document. In other words, it treats carbon dioxide emission rights as a commodity, and when carbon emissions have commodity properties, it can effectively motivate producers, investors, and enterprises to optimize their own production capacity structure and independently carry out technological innovation. Since 18 June 2013, seven pilot provinces and cities, Shenzhen, Shanghai, Beijing, Guangdong, Tianjin, Hubei, and Chongqing, have launched their carbon emissions trading policies one after another, moving towards the goal of achieving carbon peaking and carbon neutrality early. There have been many studies that have shown that the carbon emissions trading policy has profound implications for China’s carbon peaking and carbon neutrality, largely driving high energy-consuming and high-polluting industries to achieve technological breakthroughs and green low-carbon transformation; thus, having a significant effect on carbon reduction in the pilot areas [6,7]. At present, the European Union is at the forefront of promoting emissions trading, and has achieved good results, and China’s carbon emissions trading policy is of great significance in order to achieve the “double carbon” target, and this study argues that other countries that want to achieve carbon emission reduction can learn from this model and promote industry emission reduction through market measures. As the object of study for many emission reduction policies, the development and changes of enterprises at all levels are worthy of attention. Technology innovation is an important guarantee for enterprises to enhance their competitiveness and sustainable development, and the relationship between it and environmental regulation has been the focus of academic debates. Therefore, it is important for us to focus the discussion on environmental regulation at the enterprise level.

Can China’s carbon emissions trading system (ETS) influence enterprises’ green technology innovations? Is there heterogeneity? If the answer is affirmative, then this study will provide some reference for policy makers. Unlike previous approaches of using all listed enterprises in the pilot as the experimental group, the main contribution of this paper is that it takes the manually screened pilot provincial and municipal emission-controlled enterprises as the experimental group to obtain a more accurate quasi-natural experiment. The above aspects are then discussed in a comprehensive manner, with a view to developing references for deepening China’s ETS reform, and helping to achieve carbon capping and carbon neutrality.

The rest of this paper is organized as follows. Section 2 presents the literature review. Section 3 outlines the choice of model. The data used in this study are reported in Section 4. Section 5 reports the empirical results of this study, while the robustness tests are presented in Section 6. Section 7 provides a heterogeneity analysis. Section 8 concludes the paper and makes some recommendations.

## 2. Literature Review

### 2.1. Green Technology Innovation

At present, a definition of green technology innovation has not been unified, and many scholars have elaborated on suggestions attained from different perspectives. The concept was first proposed in 1994, and was defined as new processes and products that improve resource utilization efficiency and reduce environmental impact [8]. Later, scholars considered that green technology innovation was closely related to “environmental innovation” and “eco-innovation”, which is a general term for the green improvement of technology [9]. It was also argued that green technological innovation also includes green products, services, production processes, or management systems [10]. Although the existing literature differs in the definition of green technology innovations, in general, it emphasizes a focus on the concept of green development in technology innovations and achieving synergy between environmental governance and economic development.

Unlike ordinary innovation, green technology innovation is typically characterized by “double externalities” [11]. On the one hand, positive externality, which means that although enterprises developing green technologies bear most of the research and experimental development (R&D) costs, the knowledge spillover that occurs when the new technology is diffused will lead to a mismatch between the private and social benefits of innovation, and the technology developers will not be able to exclusively enjoy the benefits of innovation. On the other hand, negative externality, which means that because green technology innovation can reduce pollution emissions and resource depletion, it will generate less external environmental costs, however, enterprises tend to underestimate the cost of pollution control, which in turn leads to insufficient incentives for enterprises in this regard and ultimately reduces green innovation.

As society pays more attention to environmental problems and green development, green technology innovation has also received growing attention, and its measurement methods have become increasingly sophisticated. Some scholars have used the DEA method or the EBM model [12,13], and there have also been articles using GTIS and GTIP [14], as well as green patents [15,16].

### 2.2. Environmental Regulation and Green Technology Innovation

Environmental regulation, as a mandatory environmental management tool, can improve environmental issues. Additionally, with the increasing improvement of green technology innovation measurement methods, the research of environmental regulation and green technology innovation has gradually attracted extensive attention from academics; thus, the related literature has increased. There are currently three academic views on the relationship these two concepts.

The first type of view supports the classic “Porter hypothesis” proposed in 1995. Porter argued that the increased costs of environmental regulations would lead enterprises to improve their production processes and force them to conduct green technology innovation, thereby offsetting the costs of environmental management, increasing their profitability, and gaining a competitive advantage [17]. After Porter first described the possibility of a “win-win” outcome between environmental protection and enterprises’ competitiveness, his view was supported in many other studies [18,19,20,21,22,23]. Environmental regulation has a positive moderating effect between financial structure and green technology innovation [24]. A combination of flexible and rigid environmental regulations is more positive in promoting green technological innovation [25].

The second type of view supports conventional economics. In a study of German manufacturing enterprises, it was found that the intensity of environmental regulation had a negative effect on enterprises’ patent applications [26]. Enterprises may pay the costs of management due to the pressure of strict environmental regulation, which inhibits their R&D investment, and thus, their ability to innovate in green technologies [27]. The introduction of environmental regulation increases enterprises’ environmental expenditures and weakens their competitiveness, thereby inhibiting their technological innovation [28,29]. Some scholars have also argued that environmental regimes have a disincentive effect on resource-based industries’ green technology innovation [30].

The third type of view argues that there is a significant non-linear relationship between them. Scholars have found that, as environmental regulation is strengthened, its impact shows a significant U-shape with a promotion followed by a suppression [31,32]. Additionally, some scholars have proved this same trend based on provincial data pertaining to the Chinese industrial sector from 2005–2015, where they found a “compliance cost” on the innovation capacity of the Chinese industrial sector in the short term and an “innovation compensation” in the long term [33]; there was also an inverted-U relationship between them [34]. 

Of course, some scholars have put forward other views on the relationship. Some scholars have proposed that: different technology innovations respond differently to environmental regulations [35]; different environmental regulations have different impacts on green technological innovations [36]; and environmental regulation has different impacts on green technology innovation under different economic development conditions [3].

### 2.3. Carbon Emissions Trading Policy

With growing public concern regarding global warming, research on the carbon emissions trading market has been productive, and a systematic review of the literature shows that, as a kind of environmental regulation, the carbon emissions trading policy has received significant attention from scholars in recent years. There is a large portion of literature regarding the emission reduction performance of carbon trading, and scholars have generally agreed that the carbon trading mechanism has contributed greatly to environmental improvement in terms of reducing carbon emissions and reducing carbon intensity [37,38,39,40,41,42]. National carbon emissions trading policies are effective in recovering GDP losses [43], and there are return and volatility spillovers among the pilot regions [44]. The implementation of this policy can significantly increase the environmental investment of enterprises, which promotes the improvement of their environmental and financial performance [45,46], and improves profitability and reduces debt burden [47]. In terms of research methods, the CGE model [48] and DEA model [49] are mainly used to study carbon trading and its impacts. In addition, the synthetic control method [50], DID method [51,52,53], and PSM−DID model [54] are important methods for related studies in recent years.

### 2.4. Carbon Emissions Trading Policy and Green Technology Innovation

The existing literature mainly addresses the following features: Using the EU ETS as the object of study and finding that it promoted a 10% increase in low carbon innovation patents [55]. Using China’s ETS as a research object; one of the core objectives of carbon emissions trading policies is to promote science and technology innovation [56]. Even though it has only been a decade since China first introduced its carbon emissions trading policy, scholars have studied the relationship between both systems. The typical view is that China’s ETS significantly promotes enterprises’ green technology innovation capability [57,58]. However, some scholars have determined that the relationship was insignificant [59]. The pilot policy resulted in a decline of about 9.26% in green patents, which had a lagging effect on inhibiting corporate green innovation [60].

The literature provides some key ideas for this study. However, there are still controversies concerning the impact of ETS on green technology innovation, and no consensus view has been reached. At the same time, existing studies are less likely to explore the impact of ETS on green technology innovation of emission-controlled and non-emission-controlled enterprises, especially on the heterogeneity of the industries to which the enterprises belong. Therefore, it remains theoretically and practically significant to explore the impact of ETS on green technology innovation through the PSM−DID model and a series of heterogeneity analyses.

## 3. Methodology 

ETS in China is not random in terms of the selection of pilot regions, but may be heterogeneous with the level of green development, economic scale, and the labor force population of each province and city. Therefore, the PSM−DID method was chosen to address the problems of sample self-selection bias and heterogeneity. The PSM−DID model is a combination of propensity score matching and differences-in-differences. PSM screens control individuals from treated individuals, and DID is used to identify the effects of policy shocks.

The PSM method is used to reduce the influence of confounding variables and bias in observational studies, and eliminate confounding factors between groups for a variety of reasons, so that a more reasonable comparison can be made between experimental and control groups. In this paper, we select the listed emission-controlled enterprises in the pilot regions as the treatment group and the listed enterprises in other provinces as the control group. We then defined a two-dimensional dummy variable treat = {0, 1}, treat = 0, representing the non-emission-controlled enterprises outside the pilot, and treat = 1, representing the emission-controlled companies within the pilot regions, to estimate the propensity score based on several characteristics of the enterprises. 

The DID method is mainly used to construct a treatment group with policy implementation and a control group without implementation, and to explore the true effect of policy implementation by controlling other influencing factors. The specific econometric model is as follows:(1)$$patent=\beta_{0}+\beta_{1}D_{it}+\beta_{2}treat+\beta_{3}time+\beta_{4}\sum Controls_{it}+\gamma_{t}+\eta_{j}+\varepsilon_{it}$$
where *D_it_* is the core explained variable and its coefficient *β_1_* is a DID estimator reflecting the impact of pilot policy. *Controls_it_* are the control variables. *β_2_*, *β_3_* represent the coefficients of the dummy variables “treat” and “time”, respectively. *Controls_it_* are the control variables. *β_4_* is the coefficient of the control variable, which is meaningful for the discussion of this paper. It can explore the impact of control variables on enterprises’ green technology innovation. *η_j_* and *γ_t_* represent the dummy variables. *ε_it_* is the error term.

## 4. Data and Variables

### 4.1. Data Resource

The research objects were the listed enterprises from 2010 to 2019, and the emission-controlled enterprises in each pilot region were considered as the treatment group. Considering the availability of data, only the companies listed before 2010 were considered here. Additionally, the listed companies in non-pilot regions (non-pilot regions are provinces other than those mentioned above) were used as the control group. The list of emission-controlled enterprises in Beijing, Tianjin, Shanghai, Guangdong, Shenzhen, and Hubei were obtained from the Development and Reform Commission, while Chongqing did not publish the list of emission-controlled enterprises, so only these six regions were considered in this paper.

The number of enterprises’ green patent applications is from Chinese Research Data Services, and the data of enterprises were taken from the China Stock Market and Accounting Research Database. In addition, the following operations were performed in this paper: (1) screening out the sample of the listed companies under special treatment; (2) screening out the sample of companies in financial, educational, and comprehensive industries; (3) screening out the sample with missing values from 2010 to 2019.

### 4.2. Variable Description

In this paper, we chose 2013 as the time point of the policy shock, which is explained as follows: although the carbon emission trading policy was issued in October 2011, seven provinces and cities started ETS from June 2013. The policy was launched in the second half of 2013 and the first half of 2014, so 2013 is taken as the year of the policy implementation. When treat = 1 the sample is an emission-controlled enterprise, otherwise 0; 1 when time is greater than or equal to 2013, otherwise 0. Controls indicate control variables; this paper selected CI, ROA, COST, ROE, T1, T10, Debt, and NP [4,20,59,61,62] as control variables. The symbols and calculation methods of all variables are shown in Table 1. And the multicollinearity test is shown in Table 2.

(1)Green invention patent applications: For enterprises, ETS in China not only directly limit carbon emissions in the production process, but also have a more intuitive impact on green process innovation, and patents represent the output effect of technological innovation [11]. Considering the characteristics of invention patents and utility model patents, invention patents emphasize “outstanding substantive features” and “significant progress”, while only “substantive features and progress” are mentioned for utility model patents [41]. Naturally, the degree of inventiveness of inventions is higher than that of utility models. Therefore, the number of green invention patent applications is chosen as the explained variable.(2)CI: It can be used to indicate the size of a company and is closely related to its productivity.(3)COST: As a lubricant of technological innovation, if R&D expenses are not reasonably allocated, it is not only difficult to achieve the technological innovation goal, but also brings higher debts due to innovation. Therefore, R&D expenses are chosen as the control variable in this paper.(4)ROA: Measures the ability of the assets owned by the business to earn earnings before interest and tax (EBIT) for the business.(5)ROE: A measure of enterprises’ short-term performance. A higher ROE indicates that an enterprise can reasonably allocate the flow of capital in various production and operation areas to achieve higher technological innovation. Therefore, ROE is selected as the control variable.(6)T1/T10: The controlling shareholders of the enterprise will interfere with the technological innovation of the enterprise in the long-term plan. Shareholders provide the necessary conditions for innovation resources to the enterprise, especially those who are on the board of directors. Therefore, shareholders are bound to have an impact on the enterprise’s investment in green innovation [63]. Therefore, they are included as control variables.(7)Debt: Generally speaking, proper debt can help enterprises to invest and expand their production scale, and thus have the ability to increase green innovation. However, if it is too high, the financial risk is higher, which may lead to insufficient cash flow and a broken capital chain. Additionally, enterprises will also have difficulty in seeking investment due to insufficient solvency, and then the investment will be reduced. In summary, it is scientific and reasonable to introduce debt as a control variable in the research model.(8)NP: Enterprises that implement green technology innovation can capture markets and increase revenues by developing environmentally friendly products and technologies. By saving energy and recycling materials, they can reduce the cost of operating processes and raw materials.

## 5. Experimental Analysis

### 5.1. The Results of Propensity Score Matching

In this paper, the PSM method was chosen to match a reasonable control group for the treatment group. Caliendo and Kopeinig determined that only variables that affect both the explained variable and the sample participation decision can be chosen as matched variables [64]. Then 1:1 matching was performed with the psmatch2 command, followed by a balance test (Table 3). After matching, the absolute value of the standardized deviation of all matched variables was less than 5%, which we considered acceptable. The reason was that it has been suggested that more effective matching results can be obtained when the value is less than 20% [65]. *t*-tests with accompanying probabilities all greater than 10%, the original hypothesis was accepted, indicating that there was no significant difference between the two groups of samples on the matching variables. Therefore, the experimental and control groups after PSM matching were consistent with the common trend hypothesis, while the results of this method provided the basis for the DID model in this paper. Figure 1 depicts the kernel density before and after matching, and trend curves of the two groups are approximately overlapping, which again verifies that, with the help of PSM, the research error can be reduced and the accuracy of the DID model can be improved.

### 5.2. Regression Analysis

This paper examines whether the implementation of ETS affects enterprises’ green technology innovation by using the DID method; at the same time, selecting time and industry as fixed effects. To test the reliability of the coefficients of variables, we chose to add the control variables to the regression model in turn; the results (Table 4) were as follows:

The results show that when no control variables are included, in Model (1), the coefficient of *D_it_* is significant and positive. It indicates that the ETS policy in China significantly promotes enterprises’ green technology innovation without considering other control variables. Additionally, the coefficient of *D_it_* is still significant and positive at least at the 5% level. When the control variables are added in turn, which once again verifies the above result. This can be explained by the fact that the carbon emissions trading policy allows enterprises to trade carbon credits in the market at random, and those enterprises with high technological innovation capability can reduce carbon emissions and sell them to technologically lagging enterprises to gain benefits. Therefore, enterprises would like to increase their green technology innovation capabilities to gain benefits. Comparing the results again, it is found that the coefficient with the inclusion of all control variables is smaller than that without control variables, suggesting that environmental regulation may be overestimated if other factors are not controlled, and proving that the estimation results of the model are reliable.

It is worth mentioning that, among all the control variables, the R&D expenses (COST) has the largest positive contribution to enterprises’ green technology innovation, and is significant at the 1% level. This can be explained by the fact that appropriate environmental regulation can promote more innovative activities, and thus, increase the productivity of enterprises, offsetting the costs caused by environmental regulation, allowing them to maximize their own benefits while reducing their emissions. And then, make a trade-off between costs and controlling carbon emissions [18,19,20,21].

It is also worth mentioning that the coefficients of T1 and T10 are significantly positive. The higher the concentration of equity, the closer the relationship between shareholders’ interests and corporate interests. At this time, large shareholders will pay more attention to the supervision of managers and the enterprise’s long-term development, and then invest more in green innovation activities to promote resource allocation and enhance core competitiveness [54].

## 6. Robustness Tests

### 6.1. Parallel Trend Test

The prerequisites for using DID is that the common trend assumption must be satisfied. As can be seen in Figure 2, the coefficients are insignificant before 2013 (95% confidence interval crosses the dashed line at the level of coefficient equal to 0), while in 2015, the coefficient is significant, showing that the parallel trend assumption is satisfied. The reason why the policy has the greatest effect in the second year after implementation can be explained by the fact that invention patents are characterized by high cost, low rates, and high requirements, and therefore have higher stability for invention-based protection for the same technical method. 

### 6.2. Robustness Tests

To test the robustness of estimations (Table 5), we chose to use the following three methods: (1) Eliminating other policy’s effects. Considering the uniqueness of the policy, since the changes occurring after the policy intervention may not be caused by this policy, but by other policies during the same period. In order to exclude interference from the Twelfth Five-Year Plan, data from 2011 to 2015 were selected in this paper. The results are shown in Model (10). (2) Full-sample regression. Although the estimation of PSM−DID can match the most similar control group for the experimental group and alleviate the selectivity bias, it also loses a lot of samples. Therefore, we ran a regression on the full sample to further test its robustness, the results are presented in Model (11). (3) Replacing the explained variable. We chose the amount of authorization as the explained variable; the results are shown in Model (12). The above results demonstrate the robustness of our estimations.

## 7. Heterogeneity Analysis

In order to verify whether the results are reasonable, we conducted heterogeneity tests, including spatial heterogeneity tests, ownership heterogeneity tests, and industry heterogeneity tests.

### 7.1. Spatial Heterogeneity Tests

First, the total sample was divided into three parts according to location, and the regressions were conducted separately. When processing the data, it was found that the western region did not have the pilot provinces selected in this paper, so the results only existed for the east and middle regions. Models (15) and (16) of Table 6 show that the coefficient of *D_it_* (3.157) is significant and positive, at least at the 5% level, in the eastern region, while it (−2.710) is significant and negative in middle region. It indicates that the ETS in China significantly improved the enterprises’ green technology innovation in the eastern region, while the enterprises in the central region were inhibited. The eastern region has a developed economy, high development potential, and certain resource endowment and geographical advantages, coupled with sufficient high-quality talents and superior industrial structure, while its environmental information disclosure intensity is higher than that of the central region [66], the compensation effect of carbon trading can be reflected. While the development of the central region lags behind that of the east, the high-polluting enterprises in the east have been transferred to the central region one after another, leading to the carbon trading to produce compliance cost.

### 7.2. Ownership Heterogeneity Tests

Further examining the heterogeneity of enterprises’ ownership, the sub-samples of SOEs and non-SOEs were screened out to separately run a regression. As can be seen in Models (17) and (18) of Table 6, the coefficient of *D_it_* (1.044) is not significant in SOEs, while it (2.802) is significant and positive in non-SOEs, which indicates that the ETS policy significantly promoted the green technology innovation in non-SOEs, but had no impact on SOEs. This may be because SOEs have strong government support and easier access to green credit, and thus, have more funds to spend on innovation. During the implementation of environmental regulation policies, SOEs tend to be insensitive to external market-provided information on efficiency improvements and incentives for technological innovation [67]. Secondly, due to the agency problem in SOEs, the interests of owners and managers are not aligned, causing managers to invest in green technology innovation.

### 7.3. Industry Heterogeneity Tests 

This paper distinguished high-tech industries and non-high-tech industries, and then classified eighteen industries [68], including pharmaceutical manufacturing and special equipment manufacturing, as high-tech industries, and others as non-high-tech industries. As shown in Models (19) and (20), the coefficient (2.699) is significantly positive in non-high-tech industries, while it (0.845) is not significant in high-tech industries. Therefore, it has no impact on green technology innovation of high-tech enterprises, while it has positive impact on that of non-high-tech enterprise, which can be explained by the fact that high-tech industries involve a wide range of fields, so after the implementation of policy, its innovation involves all aspects and may not be able to consider the green innovation, while enterprises in non-high-tech industries can quickly lock in it after the policy is introduced.

## 8. Conclusions and Suggestions

Based on data of enterprises from 2010 to 2019, this paper applied the PSM−DID method to test the impact of carbon emissions trading policy on enterprises’ green technology innovation. The results show that: (1) ETS policy can significantly promote enterprises’ green technology innovation. (2) When spatial heterogeneity is considered, it is found that have a significant contribution to emission-controlled enterprises in the east, but has no effect in central enterprises. (3) When considering the heterogeneity of enterprise ownership, ETS significantly promotes the green technology innovation of non-SOEs, while having no effect on that of SOEs. (4) Considering the heterogeneity of enterprise industry, it significantly promotes the green technology innovation of non-high-tech enterprises, but has no effect on that of high-tech enterprises.

We believe that the following challenges remain for China to reduce carbon emissions in the future. (1) China’s ETS lacks a legal basis, and the market regulatory mechanism is not yet sound. (2) The coverage of ETS is narrow, and a multi-level market has not yet been formed. (3) The connection between regions is insufficient and the liquidity of the market is low. (4) Insufficient government support makes it difficult for the green technology innovation market to reach economies of scale in the short term.

According to the results, this paper makes the following recommendations:(1)Deepen carbon emissions trading policy reform and accelerate carbon trading-related legislation. On the one hand, since China’s carbon emissions present characteristics of large emissions and regional dispersion, policy coverage should be expanded and a multi-level carbon trading market should be developed. On the other hand, the upper limit of carbon prices should be set to control the emission reduction cost of enterprises, while the lower limit should serve to promote the technical emission reduction of enterprises. At the same time, China should further regulate the domestic carbon trading market, issue laws and regulations related to carbon trading, and realize a legally binding carbon trading market.(2)The government should improve the quality of green technology innovation through financial support and policy guidance. Since green technology innovation is characterized by double externalities, it needs to consider multiple goals, such as economic and social benefits, and so the government should design more incentives to innovate its supply and demand mechanism. At the same time, compared with developed countries, China has a large gap compared to them in this regard, and the problems of narrow scope and small market scale are difficult to solve by enterprise alone, which require corresponding government policy interventions and mechanism reform.(3)According to the results of this paper, non-state enterprises included in carbon emissions trading policy have stronger green technology innovation capability compared to state-owned enterprises, and non-high-tech industries are stronger compared to high-tech industries, so the government should continue to provide more policy support to non-state enterprises and non-high-tech industries in terms of green innovation, considering the heterogeneous characteristics of the industries. At the same time, it should stimulate state-owned enterprises to strengthen their motivation for green technology innovation and guide high-tech industries to focus on green technology innovation.(4)Enterprises should increase innovation investment and strengthen the construction of carbon management systems. On the other hand, enterprises should appropriately increase green R&D investment, accelerate green upgrading of enterprises, and at the same time, set reasonable carbon emission reduction targets, in order to create greater output value with less energy consumption and lower pollution. On the other hand, there should be an improvement of the financial system of carbon trading. The government should: develop financial management methods for carbon trading; strengthen the traction of carbon trading budget; incorporate carbon emission budgets into the comprehensive budget management system; and promote enterprises to save carbon and reduce costs.

## Figures and Tables

**Figure 1 ijerph-19-14325-f001:**
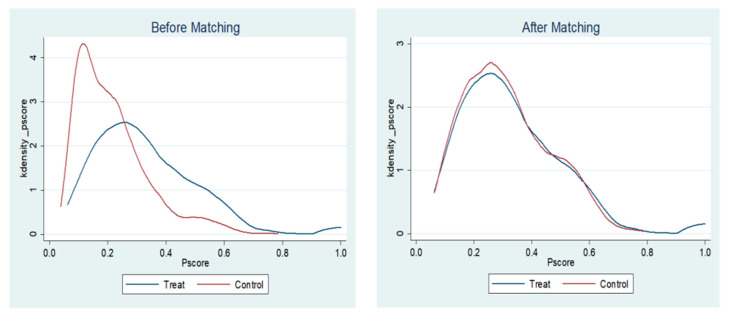
Kernel density.

**Figure 2 ijerph-19-14325-f002:**
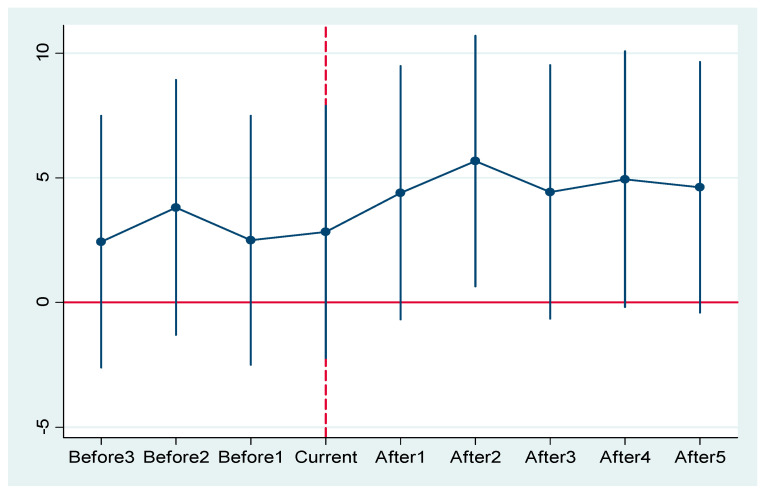
Parallel trend test.

**Table 1 ijerph-19-14325-t001:** Interpretation of variables.

Variable	Symbol	Definition/Calculation Method
Patent	Green Invention Patent Applications	Measuring the ability of enterprises’ green technology innovation
CI	Capital Intensity	Measure the size of the business
COST	R&D expenses	Ln (R&D expenses +1)
ROA	Return on Total Assets	Net profit/Average total assets
ROE	Return on Equity	Net Profit/ Net Assets
T1	The single largest shareholder	Indicate the equity concentration
T10	The largest top ten shareholders	Indicate the equity concentration
Debt	Asset-liability ratio	Used to measure corporate liabilities
NP	Net Profit	Ln (Net Profit)
Industry	Industrial dummy variable	Virtual variable
Year	Time dummy variable	Virtual variable

**Table 2 ijerph-19-14325-t002:** Multicollinearity test.

Variable	VIF	1/VIF
CI	1.154	0.867
COST	1.072	0.933
ROA	3.996	0.25
ROE	3.087	0.324
T1	1.854	0.539
T10	1.836	0.545
Debt	1.516	0.66
NP	1.166	0.857

**Table 3 ijerph-19-14325-t003:** The PSM validity test.

Variable	Unmatched	Mean		%Reduct		*t*-Test	
	Matched	Treated	Control	%Bias	Bias	t	*p* > t
CI	U	3.937	2.206	14.400		4.650	0.000
	M	2.153	2.104	0.400	97.400	0.580	0.562
ROE	U	0.066	0.081	−9.500		−2.540	0.000
	M	0.072	0.073	−0.400	95.900	−0.7	0.941
T1	U	40.887	32.983	51.300		12.270	0.000
	M	39.888	40.446	−3.600	92.900	−0.65	0.516
Debt	U	0.497	0.393	53.500		12.620	0.000
	M	0.486	0.487	−0.100	99.800	−0.02	0.982

**Table 4 ijerph-19-14325-t004:** Benchmark regression results.

	(1)	(2)	(3)	(4)	(5)	(6)	(7)	(8)	(9)
VARIABLES	Patent	Patent	Patent	Patent	Patent	Patent	Patent	Patent	Patent
D_it_	3.579 ***	3.516 ***	4.362 ***	4.389 ***	4.399 ***	4.390 ***	4.112 ***	4.121 ***	2.183 **
	(1.168)	(1.164)	(1.176)	(1.175)	(1.175)	(1.161)	(1.161)	(1.161)	(1.028)
CI		−1.083 ***	−0.814 **	−0.924 ***	−0.903 ***	−0.746 **	−0.781 **	−0.683 *	−0.462
		(0.333)	(0.338)	(0.343)	(0.343)	(0.340)	(0.339)	(0.348)	(0.307)
COST			0.817 ***	0.828 ***	0.841 ***	0.788 ***	0.807 ***	0.827 ***	0.926 ***
			(0.205)	(0.205)	(0.205)	(0.203)	(0.202)	(0.203)	(0.179)
ROA				−18.59 *	−32.97 **	−47.54 ***	−51.19 ***	−38.80 **	−35.70 **
				(9.526)	(15.32)	(15.34)	(15.35)	(18.28)	(16.10)
ROE					6.483	8.172	8.838 *	6.414	3.439
					(5.409)	(5.350)	(5.340)	(5.682)	(5.007)
T1						0.185 ***	0.109 ***	0.110 ***	0.0156
						(0.0320)	(0.0414)	(0.0415)	(0.0369)
T10							0.120 ***	0.117 ***	−0.0205
							(0.0417)	(0.0417)	(0.0374)
Debt								3.815	3.570
								(3.059)	(2.695)
NP									0.00840 ***
									(0.000436)
Year fixed effects Industry fixed effects	Y Y	Y Y	Y Y	Y Y	Y Y	Y Y	Y Y	Y Y	Y Y
Constant	2.381 ***	4.707 ***	−1.850	−0.927	−0.928	−7.825 ***	−11.92 ***	−14.38 ***	−4.243
	(0.634)	(0.955)	(1.902)	(1.958)	(1.957)	(2.272)	(2.678)	(3.323)	(2.974)
Observations	1302	1302	1302	1302	1302	1302	1302	1302	1302

Note: ***, **, and * indicate significance at the 1%, 5%, and 10% levels, respectively. Standard deviations are show in parentheses.

**Table 5 ijerph-19-14325-t005:** Results of robustness tests.

	(10)	(11)	(12)
VARIABLES	Eliminating Other Policy’s Effects	Full-Sample Regression	Replacing the Explained Variable
D_it_	2.770 **	2.847 ***	2.029 *
	(1.305)	(0.770)	(1.082)
CI	−0.553	−0.0849 ***	−0.554 *
	(0.415)	(0.0287)	(0.312)
COST	0.848 ***	1.098 ***	0.614 ***
	(0.210)	(0.137)	(0.189)
ROA	−29.03	−10.47	−26.21
	(32.41)	(8.824)	(19.54)
ROE	3.367	0.758	1.974
	(17.46)	(2.924)	(8.250)
T1	−0.0198	0.0103	2.563
	(0.0431)	(0.0234)	(2.862)
T10	−0.0266	0.00882	−0.169 ***
	(0.0446)	(0.0235)	(0.0195)
Debt	4.162	3.943 **	1.139 ***
	(3.686)	(1.600)	(0.0713)
NP	0.00826 ***	0.00689 ***	0.00205 ***
	(0.000513)	(0.000222)	(0.000630)
Constant	−2.550	−8.683 ***	−3.199
	(3.582)	(1.703)	(2.587)
Industry fixed effects Year fixed effects	Y Y	Y Y	Y Y
Observations	656	2790	1168
R-squared	0.331	0.305	0.379

Note: ***, **, and * indicate significance at the 1%, 5%, and 10% levels, respectively. Standard deviations are show in parentheses.

**Table 6 ijerph-19-14325-t006:** Results of heterogeneity test.

	(15)	(16)	(17)	(18)	(19)	(20)
VARIABLES	East	Mid	SOEs	Non-SOEs	Non-High-Tech	High-Tech
D_it_	3.157 **	−2.710 **	1.044	2.802 ***	2.699 *	0.845
	(1.336)	(1.110)	(1.881)	(1.073)	(1.501)	(1.424)
CI	−0.509	−0.281	−0.669	−0.236	−0.744 *	−0.128
	(0.416)	(0.335)	(0.529)	(0.325)	(0.406)	(0.436)
COST	1.073 ***	0.237	0.618 **	1.218 ***	0.602 ***	1.652 ***
	(0.228)	(0.181)	(0.275)	(0.248)	(0.224)	(0.403)
ROA	−39.81 **	−30.53 *	−70.80 *	−30.22 *	−93.86 ***	−29.69 *
	(19.89)	(15.94)	(38.42)	(15.42)	(30.98)	(17.70)
ROE	1.448	−1.213	4.133	−1.334	18.13*	−3.712
	(6.656)	(4.277)	(17.61)	(4.229)	(9.973)	(5.094)
T1	0.0404	−0.0611 *	0.0663	−0.0508	0.102 **	−0.144 **
	(0.0474)	(0.0335)	(0.0625)	(0.0420)	(0.0470)	(0.0606)
T10	−0.0578	0.0519	−0.0418	0.0460	−0.0402	0.0713
	(0.0501)	(0.0355)	(0.0596)	(0.0474)	(0.0498)	(0.0630)
Debt	5.547	0.264	−4.313	8.000 ***	−4.893	8.642 **
	(3.454)	(2.668)	(5.546)	(2.750)	(4.313)	(3.564)
NP	0.00838 ***	0.0459 ***	0.00837 ***	0.0377 ***	0.00805 ***	0.0436 ***
	(0.000502)	(0.00461)	(0.000572)	(0.00744)	(0.000490)	(0.00633)
Constant	−4.652	0.136	3.127	−11.03 ***	1.330	−12.22 ***
	(3.821)	(2.650)	(5.262)	(3.443)	(4.210)	(4.255)
Industry Year	Y Y	Y Y	Y Y	Y Y	Y Y	Y Y
Observations	991	212	651	651	812	490
R-squared	0.290	0.420	0.337	0.129	0.347	0.191

Note: ***, **, and * indicate significance at the 1%, 5%, and 10% levels, respectively. Standard deviations are show in parentheses.

## Data Availability

The datasets used in this study are available from the corresponding author on reasonable request.

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
