# Peer review of "The Carbon Emissions Trading Policy of China: Does It Really Promote the Enterprises’ Green Technology Innovations?"

_ijerph, 2022, doi:10.3390/ijerph192114325_

Round 1

Reviewer 1 Report

The research provided answers to the researchers' questions, which they also posed in the title of the paper. The contribution of this article is significant because it deals with a problem at the level of a large economy such as China. Weaknesses in the work refer to insufficient connection with the experiences of other countries with similar problems and potential solutions. Comments are in uploaded version.

Author Response

Dear Reviewer,

Thank you for your time spent in reading and reviewing our paper and providing these comments for us. We have made every effort to address these issues. Please see the new version of the manuscript and read our point-by-point responses for details in the “Response document”.

Thank you for your work.

Best wishes,

Yours sincerely,

Chao Feng, Ph.D.

Reviewer 2 Report

Examining the impact of the emissions trading policy on innovation is important, especially if the results of the research can be incorporated into the decision-making mechanism and regulation can be made more efficient and carbon dioxide emissions can be reduced.

It is nice that the authors examined the practical impact of China's emissions trading policy in the light of international emissions trading experiences. I consider the methodology chosen for the study (sampling, data used, selected variables and analysis methods) to be good. The reliability of the analysis and the drawing of conclusions were aided by both robustness and ownership and industry heterogeneity studies. I accept the conclusions drawn from the results of the analyzes and the suggestions made by them.

·        At the same time, it is necessary to provide the interpretation of abbreviations occurring in the text. Readers less familiar with econometric methods will find it difficult to navigate the jungle of abbreviations.

·        It would be good to provide an interpretation of the PSM-DID models.

·        n Figure 1, please also provide the interpretation of the beta coefficients.

Author Response

(The authors gave the same response as above.)

Reviewer 3 Report

The manuscript with the title “The carbon emissions trading policy of China: Does it really 2 promote the enterprises’ green technology innovations” focused on the carbon emissions trading policy of China and various factors that can affect the policy. The authors have presented the carbon emission data from seven different regions in China.

Below are my comments:

1.      Table 1, The CO2 emission in Guangdong and Hubei have increased as per the data. The policy is not working effectively in these regions as we compare it with other regions reported here. Please comments.

2.      Table 2, Which variable has a significant effect on carbon emission policy? Further, the cost is a great factor to work these factors smoothly. Is it possible to make trade-off between cost and emission control?

3.      Are these policies effective in achieving Carbon neutrality by 2060 in China? Comment please.

4.      Please write the full form of abbreviations like EBIT, ETS, CNRDS, etc. mentioned in the manuscript.

5.      If possible, please include the future challenges for reducing carbon emissions.

Author Response

(The authors gave the same response as above.)

Reviewer 4 Report

This article uses the PSM-DID model to examine the impact of carbon emissions trading policy on enterprises' green technology innovations. This study is well organized and has valid conclusions, but there are still several problems.

1. In line 27 and line 45, the format of literature annotation is wrong or inconsistent, like [1214, 19] and (Lv and Bai, 2021); Line 345 should be Table 7. Please check the full text for similar errors.

2. Line 49, ETS appears for the first time, what does the term "ETS" stand for? emissions trading system?

3. The literature review should summarize the gaps in existing research and highlight how this paper fills the research gap.

4. In Section 4.2, corresponding literature support can be added for the selection of each control variable. The statement in line 218-230 is too verbose. In addition, this paper studies the impact of carbon emission policies on green technology innovation, so the green patent application should be the explained variable Y instead of the explanatory variable X, is the statement in line 230 incorrect?

5. In Table 4, why these matched variables are chosen? What about the other variables? Please explain in the text.

6. In section 7, Heterogeneity analysis is the main source of conclusions and should be closely linked to specific data, just a sentence “As shown in Models” is not accurate enough.

7. In which non-pilot area are he data of the control group collected from? please specify.

Author Response

(The authors gave the same response as above.)
